

# Effect of common and experimental anti-tuberculosis treatments on *Mycobacterium tuberculosis* growing as biofilms

James P. Dalton[1,2,3], Benedict Uy[1,2], Narisa Phummarin[4], Brent R. Copp[3,4], William A. Denny[3,5], Simon Swift[1] and Siouxsie Wiles[1,2,3]

[1] Molecular Medicine and Pathology, University of Auckland, Auckland, New Zealand
[2] Bioluminescent Superbugs Lab, University of Auckland, Auckland, New Zealand
[3] Maurice Wilkins Centre for Molecular Biodiscovery, Auckland, New Zealand
[4] School of Chemical Sciences, University of Auckland, Auckland, New Zealand
[5] Auckland Cancer Society Research Centre, University of Auckland, Auckland, New Zealand

## ABSTRACT

Much is known regarding the antibiotic susceptibility of planktonic cultures of *Mycobacterium tuberculosis*, the bacterium responsible for the lung disease tuberculosis (TB). As planktonically-grown *M. tuberculosis* are unlikely to be entirely representative of the bacterium during infection, we set out to determine how effective a range of anti-mycobacterial treatments were against *M. tuberculosis* growing as a biofilm, a bacterial phenotype known to be more resistant to antibiotic treatment. Light levels from bioluminescently-labelled *M. tuberculosis* H37Rv (strain BSG001) were used as a surrogate for bacterial viability, and were monitored before and after one week of treatment. After treatment, biofilms were disrupted, washed and inoculated into fresh broth and plated onto solid media to rescue any surviving bacteria. We found that in this phenotypic state *M. tuberculosis* was resistant to the majority of the compounds tested. Minimum inhibitory concentrations (MICs) increased by 20-fold to greater than 1,000-fold, underlying the potential of this phenotype to cause significant problems during treatment.

# INTRODUCTION

The bacterium *Mycobacterium tuberculosis* is responsible for the lung disease tuberculosis (TB). It is estimated that one-third of the world's population is infected with this deadly pathogen (*World Health Organization, 2015*). While TB represents a huge burden on health care systems in its own right, it also complicates other serious illnesses such as HIV/AIDS (*World Health Organization, 2015*). New compounds are desperately required to shorten the current TB treatment regimes, which routinely last longer than 6 months, and to combat the rise of resistant *M. tuberculosis* strains. Drug-resistant TB leads to extended hospital stays and treatment times and, in the worst case scenario, untreatable disease (*Klopper et al., 2013*; *World Health Organization, 2011*).

Corresponding author
Siouxsie Wiles,
s.wiles@auckland.ac.nz

Bacterial cells present in an infected host can display a range of phenotypes and occupy several divergent physiological niches (*Sendi et al., 2009*; *Tuchscherr et al., 2011*). For example, during infection, cells of *M. tuberculosis* can be both replicative and non-replicative (*Wayne & Sohaskey, 2001*) and occupy a number of different niches, including macrophages (*Rengarajan, Bloom & Rubin, 2005*) and necrotic and non-necrotic lesions (*Fenhalls et al., 2002*). *M. tuberculosis* growing in such diverse environments is unlikely to be accurately reflected by planktonically-grown laboratory cultures. Many bacteria can form microcolonies called biofilms, which can contain a mixture of replicating and non-replicating cells, and cells in different metabolic states (*Rice, Hamilton & Camper, 2003*; *Stewart & Franklin, 2008*). Bacteria can form biofilms at the interface between a surface and the surrounding air or liquid. Alternatively, floating biofilms can form at a liquid/air interface. These floating biofilms are also known as pellicles. Within a biofilm or pellicle, bacterial cells are more resistant to disinfection and drug treatment and therefore represent a much harder target to sterilise (*Kulka, Hatfull & Ojha, 2012*; *Ceri et al., 1999*). As such, the biofilm/pellicle represents a useful model for investigating the efficacy of antibacterial treatments.

*M. tuberculosis* can form pellicles *in vitro* (*Sambandan et al., 2013*) and the presence of microcolonies of extracellular *M. tuberculosis* in animal models has led to speculation that these are biofilms formed *in vivo* (*Lenaerts et al., 2007*). Some sources point to the possible presence of pellicles in the lung-air interface present in secondary TB in humans; (*Hunter et al., 2006*; *Hunter et al., 2013*; *Hunter et al., 2014*) and indicate that the susceptibility of this phenotype to antibacterial compounds is of particular relevance from a treatment standpoint. Here we describe the use of bioluminescently-tagged *M. tuberculosis* to investigate the susceptibility of pellicle cells to a range of anti-mycobacterial compounds, including those in current clinical use as well as a selection of experimental compounds.

## MATERIALS AND METHODS

### Strains and growth conditions

In this study we used *M. tuberculosis* BSG001 (*Wang et al., 2016*), a stable bioluminescent derivative of H37Rv transformed with the integrating plasmid pMV306hspLuxABG13CDE (*Andreu et al., 2010*). Cultures of BSG001 were grown at 37 °C with gentle shaking (100 rpm) in Middlebrook 7H9 broth (Fort Richard, Auckland) supplemented with 10% Middlebrook ADC enrichment media (Fort Richard) and 0.5% glycerol (Sigma-Aldrich), or on 7H11 agar (Fort Richard) supplemented with 10% Middlebrook OADC enrichment media (Fort Richard) and 0.5% glycerol. We grew pellicles in sterile, black 96 well plates (Nunc) using a previously described method (*Kulka, Hatfull & Ojha, 2012*). Briefly, we grew *M. tuberculosis* BSG001 in liquid culture for 2 weeks at 37 °C and then adjusted the cultures to give an optical density at 600 nm ($OD_{600}$) of 1.0, before diluting them 1:100 in modified Sauton's media (0.5 g/L $KH_2PO_4$, 0.5 g/L $MgSO_4$, 4 g/L L-Asparagine, 2 g/L Citric acid, 0.05 g/L Ferric Ammonium Citrate, 60 mL/L glycerol, 0.1% $ZnSO_4$, pH 7.0 (all chemicals from Sigma-Aldrich)) and adding 100 μL aliquots to each inner well of a 96

**Table 1** Minimum inhibitory and bactericidal concentrations of common and experimental anti-tuberculosis treatments against *Mycobacterium tuberculosis*.

| | Planktonic MIC[a] | Biofilm MIC[a] | Planktonic MBC[b] | Biofilm MBC[b] |
|---|---|---|---|---|
| Pyrazinamide | 50 mg/L | >1,000 mg/L | 50 mg/L | >1,000 mg/L |
| Rifampicin | 0.04 mg/L | 4 mg/L | 0.04 mg/L | 4 mg/L |
| Isoniazid | 0.04 mg/L | >256 mg/L | 0.08 mg/L | >256 mg/L |
| Ethambutol | 1 mg/L | >2,000 mg/L | 2 mg/L | >2,000 mg/L |
| Streptomycin | 0.5 mg/L | 125 mg/L | 0.5 mg/L | 1,000 mg/L |
| Amikacin | 4 mg/L | 250 mg/L | 4 mg/L | 1,000 mg/L |
| Rifabutin | 0.04 mg/L | 8 mg/L | 0.04 mg/L | 16 mg/L |
| Ascorbic acid | 700 mg/L | 2,800 mg/L | 700 mg/L | 2,800 mg/L |
| Delamanid | 0.042 mg/L | >53.45 mg/L | 0.042 mg/L | >53.45 mg/L |
| Pretomanid | 0.011 mg/L | >3.6 mg/L | 0.011 mg/L | >3.6 mg/L |
| SN30488 | 0.0056 mg/L | >4.2 mg/L | 0.0056 mg/L | >4.2 mg/L |
| QOA1 | 0.5 mg/L | >128 mg/L | 0.5 mg/L | >128 mg/L |
| QOA2 | 0.25 mg/L | >128 mg/L | 0.25 mg/L | >128 mg/L |
| QOA3 | 0.25 mg/L | >128 mg/L | 0.25 mg/L | >128 mg/L |
| 5-FAA[c] | 9.7 mg/L | 19.4 mg/L | 19.4 mg/L | 19.4 mg/L |
| 6-FAA[c] | 9.7 mg/L | 19.4 mg/L | 19.4 mg/L | 19.4 mg/L |

**Notes.**
[a] Minimum inhibitory concentrations (MIC).
[b] Minimum bactericidal concentrations (MBC) for biofilm and planktonic forms *of M. tuberculosis* BSG001 for a variety of experimental and non-experimental compounds.
[c] Fluoroanthranilic acid.

well plate. We filled the outer wells with 200 µL of sterile water to reduce evaporation from the *M. tuberculosis* containing wells. We incubated the cultures without shaking for 8 weeks at 37 °C.

## Determination of compound activity

Once pellicles had established, we determined how much media remained in the wells by removing all liquid from 12 non-tested wells and taking an average volume. This is necessary as some evaporation occurs because of the long incubation time, and this has to be accounted for when calculating final compound concentrations. Similar levels of liquid were lost from all wells tested. We added test compounds (Table 1 & Fig. 1, all supplied by Sigma Aldrich with the exception of the nitroimidazole and 2-(quinoline-4-yloxy) acetamide-based compounds) dissolved in Sauton's media to the *M. tuberculosis* pellicles in a two-fold dilution gradient at a range of concentrations. Concentrations of each compound were chosen based on the minimum inhibitory concentrations (MIC) for planktonically-grown *M. tuberculosis* BSG001, which vary greatly between the compounds tested. We tested up to 1,000-fold the planktonic MIC concentration, depending on the solubility of the test compound. Each concentration was done in duplicate, using three independent BSG001 cultures. Biofilms were incubated for a further 7 days at 37 °C with no shaking and light levels (given as Relative Light Units [RLU]) were measured before and after treatment using a Victor X-3 luminometer (Perkin Elmer). We have

**Figure 1** Chemical structures of the experimental compounds used in this study.

defined the MIC as causing a 1 log reduction in light production, as previously described (*Andreu et al., 2012*).

To determine the minimum bactericidal concentration (MBC), pellicles were removed from the wells, disrupted by pipetting and washed 3 times in Sauton's media supplemented with 0.05% tween 80. The cells were then resuspended in fresh 7H9 broth (5 ml, supplemented as described above) and plated onto 7H11 agar (supplemented as described above). We have defined the MBC as the concentration which resulted in no growth. We incubated broth cultures for 2 weeks and plate cultures for eight weeks to recover any surviving bacteria. All experiments were performed using three biological replicates of *M. tuberculosis* BSG001 and two technical replicates. Biological replicates were grown separately and tested on different days.

## RESULTS

### Decreased susceptibility of pellicle-grown *M. tuberculosis* to front-line and experimental compounds

Of the four main first line drugs only rifampicin was seen to inhibit pellicles of *M. tuberculosis* at the concentrations tested (Table 1, Figs. 2C and 3B). Isoniazid also led to some inhibition but below the threshold defined (Figs 2C and 3C). In the case of rifampicin the MIC and MBC for pellicle-grown BSG001 were determined to be 4 mg/L, 100 times the concentration required to produce a similar result with planktonic cells (Table 1). High levels of pyrazinamide, isoniazid and ethambutol (20, 6,000 and 1,000 times the MIC's for planktonic cells, respectively) failed to sufficiently reduce light to be classed as inhibitory (Table 1, Figs. 2A–2D and 3A–3D). More success was observed with non-first line antibiotics, with MICs obtained for pellicle-grown BSG001 for streptomycin (125 mg/L), amikacin (250 mg/L) and rifabutin (8 mg/L) (Table 1, Figs. 2E–2G and 3E–3G). These pellicle-MICs represent an increase of 250, 62.5 and 200 times the MIC's obtained for planktonic cultures, respectively (Table 1).

When novel and experimental compounds were examined, none of the current derivations of the nitroimidazole based compounds (*Olaru et al., 2015*; *Palmer et al., 2010*) (delamanid, pretomanid and SN30488) were able to reduce light from the pellicle-grown *M. tuberculosis* at the concentrations tested (Table 1, Figs. 2H–2J and 3H–3J). The same resistance to drug-killing was seen with experimental compounds based on 2-(quinoline-4-yloxy) acetamides (*Phummarin et al., 2016*) (QOA 1, QOA 2 and QOA 3) (Table 1, Figs. 2K–2M and 3K–3M). In contrast, the fluoroanthranilic-acid based compounds, 5-fluoroanthranilic acid (5-FAA) and 6-fluoroanthranilic acid (6-FAA), which target the tryptophan biosynthetic pathway (*Toyn et al, 2000*) were seen to be quite effective at inhibiting light from *M. tuberculosis* BCG001 growing as a pellicles (Table 1, Figs. 2N/O and 3N/O). Ascorbic acid was also seen to cause inhibition at 2.8 g/L, 4 times the MIC for planktonically-grown *M. tuberculosis* (Table 1, Figs. 2P and 3P).

## DISCUSSION

Many infectious bacteria form biofilms within their host. Bacteria living within a biofilm are notoriously difficult to treat and can persist for extended periods of time, as they have the ability to resist the immune system (*Donlan & Costerton, 2002*), display increased virulence (*Safadi et al., 2012*; *Wand et al., 2012*) and can become phenotypically more resistant to antibiotics. It is common for antibiotic concentrations required to control bacteria within biofilms to be 100–1,000 fold greater than those needed to treat planktonic forms (*Ceri et al., 1999*). This was seen to be true of the majority of compounds tested in this study. In many cases this is not too surprising. Biofilms can affect drug activity by acting as a physical barrier to entry into the cell. The phenotypic state of the cells within the biofilm could also make the cells less susceptible. Isoniazid's mode of action relates to mycolic acid synthesis. When growing as a pellicle, it is possible that mycolic acid synthesis is minimal or nonessential. Similarly ethambutol, delamanid and pretomanid are also thought to affect various steps in cell wall biosynthesis and formation; none of these were seen to

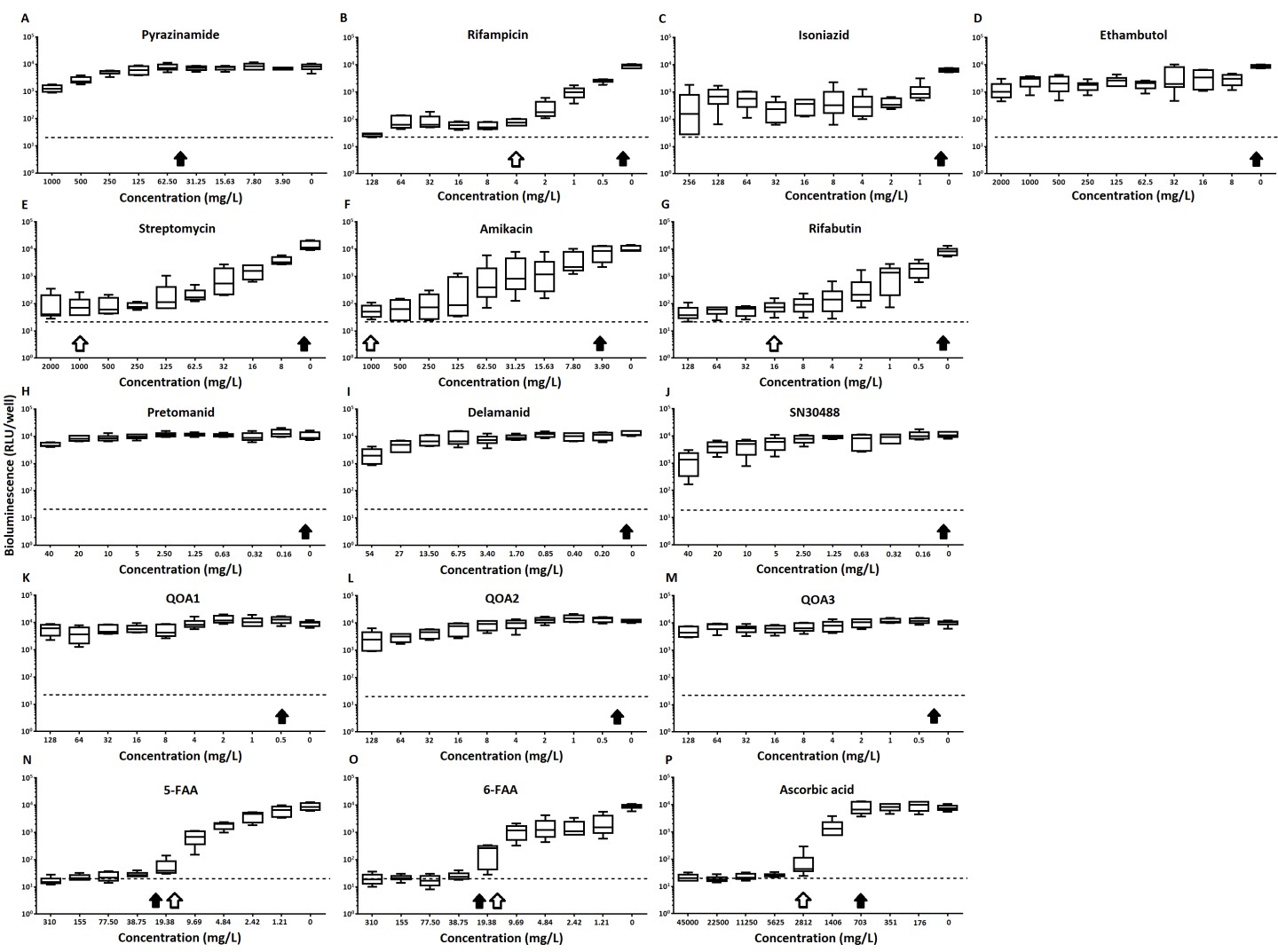

**Figure 2** **The effect of clinically-used and experimental compounds on *M. tuberculosis* BSG001 pellicles.** The inhibitory effect of first line (A–D) and second line (E–G) anti-tuberculosis drugs used in the clinic and experimental compounds (H–P), including those based on nitroimidazole (H–J), 2-(quinoline-4-yloxy) acetamides (K–M) and fluoroanthranilic-acid (N, O), is presented as a reduction in bioluminescence plotted as relative light units (RLU) per well on day 7 of treatment. The dashed line indicates the limit of detection. The solid and open arrows indicate the MBC's (the concentration which resulted in the recovery of no bacterial colonies) obtained for planktonically-grown cells and pellicles, respectively. All compounds were tested in three biological replicates on separate days with multiple technical replicates. Results are given as box whisker plots with the box representing values from the lower to upper quartile and the whiskers representing the range.

have an effect. Pyrazinamide relies on conversion to pyrazinoic acid, which requires acidic conditions to become active. This could indicate that these conditions are not present in mycobacterial pellicles or that the drug is unable to penetrate the cells in this phenotype. If it is due to the lack of an acidic environment this could represent a limitation in using this model for drug testing. In contrast, antibiotics that affect protein synthesis, such as rifampicin, rifabutin, amikacin and streptomycin, displayed some degree of inhibitive activity towards pellicle-grown *M. tuberculosis*, although this activity was lower than the activity against planktonically-grown cells. The 2-(quinoline-4-yloxy) acetamide based

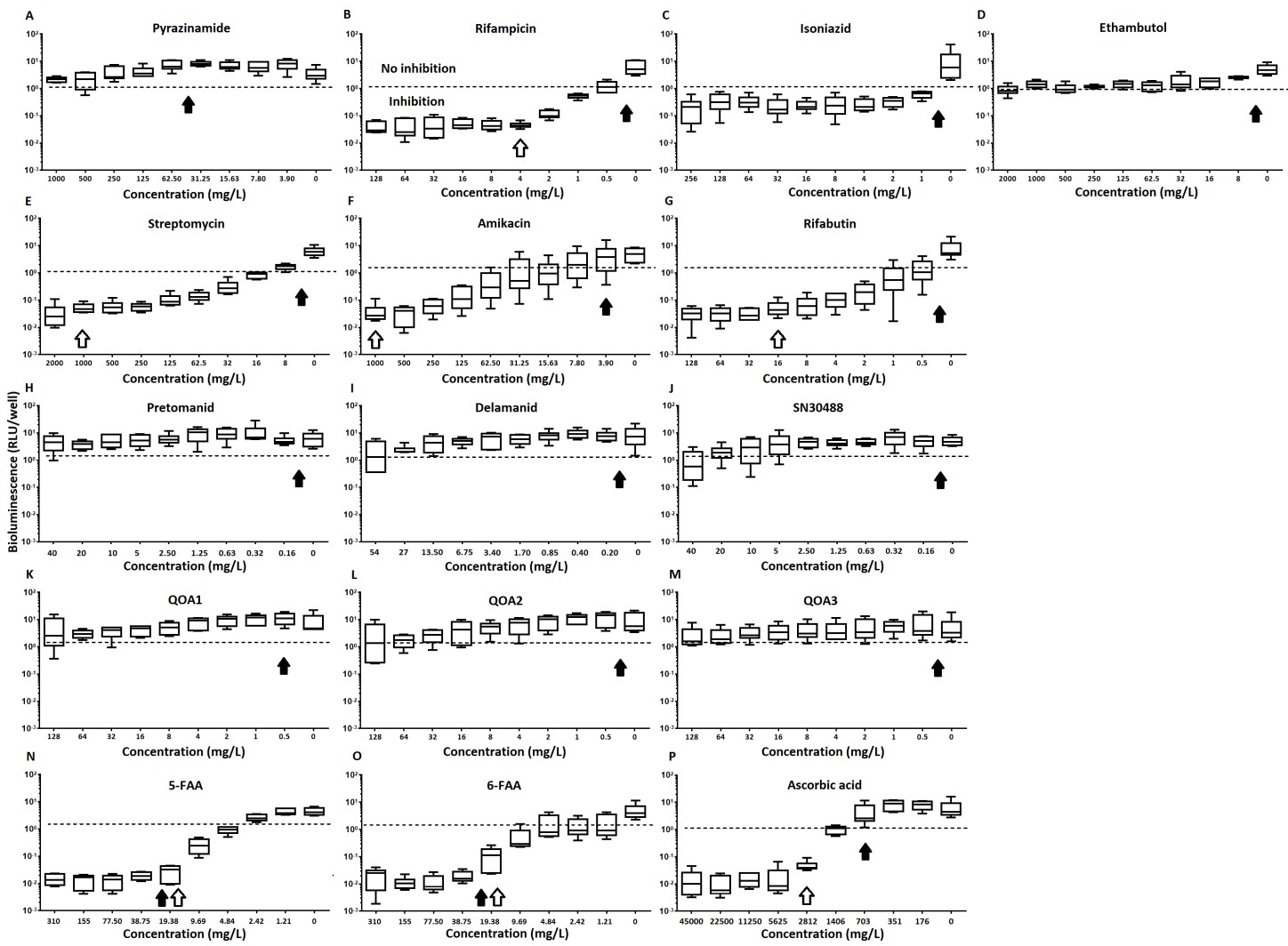

**Figure 3** **The relative effect of clinically-used and experimental compounds on *M. tuberculosis* BSG001 pellicles.** The relative change in bioluminescence (relative light units [RLU]) following the treatment *of M. tuberculosis* BSG001 biofilms with first line (A–D) and second line (E–G) anti-tuberculosis drugs used in the clinic and experimental compounds (H–P), including those based on nitroimidazole (H–J), 2-(quinoline-4-yloxy) acetamides (K–M) and fluoroanthranilic-acid (N, O), is shown as the ratio of RLU before treatment and RLU after seven days of treatment. The dashed line indicates the level at which no change occurs; values above the dashed line indicate an increase in light levels (and hence survival/-growth) over the course of the experiment, while those below indicate a decrease (and hence inhibition/death). The solid and open arrows indicate the MBC's (the concentration which resulted in the recovery of no bacterial colonies) obtained for planktonically-grown cells and pellicles, respectively. Results are given as box whisker plots with the box representing values from the lower to upper quartile and the whiskers representing the range.

compounds also showed little activity against this biofilm form. The mode of action of these compounds is likely due to electron transport inhibition of cytochrome bc1 oxidase (*Phummarin et al., 2016*). As the cells are actively metabolising, the lack of an effect from these compounds is most likely due to an inability to access the cells.

In our study, we observed that *M. tuberculosis* growing as a pellicle is susceptible to a concentration of ascorbic acid similar to that reported for planktonically-grown cells. This concentration was also sufficient to cause death of the *M. tuberculosis* pellicle within one

week of treatment. The activity of ascorbic acid is thought to be due to the generation of highly reactive hydroxyl radicals via presence of iron and Fenton reaction chemistry (*Vilcheze et al., 2013*). Killing due to this mechanism would be non-specific and not dependant on uptake. Interestingly the fluoroanthranilic acid tryptophan biosynthesis inhibitors were also seen to be effective at inhibiting and killing pellicles, indicating that this is a pathway worthy of further consideration for drug targeting.

It is possible that the comparative ease in which test compounds can access bacteria within a pellicle, that is from both above and below, as compared to a biofilm attached to a surface which cannot be accessed from the surface side, make this form of biofilm easier to kill. While it is still unknown if *M. tuberculosis* forms biofilms/pellicles *in vivo*, many mycobacterial species do form complex, secondary structures such as pellicles *in vitro* (*Ojha & Hatfull, 2012*). Researchers have also reported histological evidence for the presence of multicellular structures involving *M. tuberculosis* outside of the macrophage (*Lenaerts et al., 2007*). Others have reported the presence of cells that resembles biofilms/pellicles in the cavities formed during secondary tuberculosis which would indicate that this phenotype is likely to play a role in human disease (*Lenaerts et al., 2007*; *Hunter et al., 2006*; *Hunter et al., 2014*). The biphasic response of *M. tuberculosis* infections, in which a large kill is seen early on in drug treatment followed by a marked reduction in the bactericidal activity of therapeutic agents due to phenotypic rather than genetic resistance, could also be evidence that *M. tuberculosis* is able to form biofilms/pellicles *in vivo*. Such structures could act as a reservoir for drug tolerant bacilli which are responsible for the increased duration of drug treatment required in cases of TB. Regardless, the *M. tuberculosis*-pellicle model is a useful multi-phenotypic environment in which a novel compound can be tested against cells with a range of susceptibilities. The susceptibility of *M. tuberculosis* within this model indicates that drugs which can attack the surface of the cell or can pass through the extracellular matrix of the pellicle represent the best option for treatment. We also saw that the inhibition of tryptophan biosynthesis could be utilised in TB treatment and their design should be further investigated.

### Funding

This work was supported by the Maurice Wilkins Centre for Molecular Biodiscovery and the University of Auckland Vice-Chancellor's Strategic Development Fund and Faculty Research Development Fund. SW is supported by a Sir Charles Hercus Fellowship (09/099) from the Health Research Council of New Zealand. The funders had no role in study design, data collection and analysis, decision to publish, or preparation of the manuscript.

### Grant Disclosures

The following grant information was disclosed by the authors:
Maurice Wilkins Centre for Molecular Biodiscovery.
University of Auckland Vice-Chancellor's Strategic Development Fund.

Faculty Research Development Fund.
Sir Charles Hercus Fellowship: 09/099.

## Competing Interests

Siouxsie Wiles is an Academic Editor for PeerJ.

## Author Contributions

- James P. Dalton conceived and designed the experiments, performed the experiments, analyzed the data, contributed reagents/materials/analysis tools, wrote the paper, prepared figures and/or tables, reviewed drafts of the paper.
- Benedict Uy performed the experiments, contributed reagents/materials/analysis tools, reviewed drafts of the paper.
- Narisa Phummarin contributed reagents/materials/analysis tools.
- Brent R. Copp and William A. Denny contributed reagents/materials/analysis tools, reviewed drafts of the paper.
- Simon Swift conceived and designed the experiments, analyzed the data, contributed reagents/materials/analysis tools, reviewed drafts of the paper.
- Siouxsie Wiles conceived and designed the experiments, analyzed the data, contributed reagents/materials/analysis tools, wrote the paper, prepared figures and/or tables, reviewed drafts of the paper.

## Data Availability

Wiles, Siouxsie; Dalton, James; Uy, Benedict; Copp, Brent; Denny, Bill (2016): Effect of common and experimental anti-tuberculosis treatments on *Mycobacterium tuberculosis* growing as biofilms. figshare: 10.17608/k6.auckland.4097772.v1.

## Supplemental Information

Supplemental information for this article can be found online at http://dx.doi.org/10.7717/peerj.2717#supplemental-information.

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
