# Peer review of "Effect of common and experimental anti-tuberculosis treatments on Mycobacterium tuberculosis growing as biofilms"

_PeerJ, doi:10.7717/peerj.2717_

## Round 0.1 · original submission · Major Revisions

As you can see, your paper got mixed feelings from the reviewers. While all 3 found the study of potential interest, all 3 also identified several weaknesses that, collectively, make the paper unacceptable under its present format. Beyond the scientific issues raised, it also seems that the paper was not correctly proof-read before submission, which denotes a lack of attention. Lastly, the significance of your data in terms of future developments of chemotherapy of tuberculosis seems thin.

Reviewer 1 ·

Basic reporting

Basic reporting appears to conform to journal format and in a language that is consistent with scientific reports.

Experimental design

The experimental design has some significant shortcomings. First, the antimycobacterial activity of Vitamin C has been published. Second, even if authors claim novelty in testing the sterilization activity of Vitamin C against mycobacterial biofilms, then authors should have included additional assays. Luciferase-based luminescence assays is not sensitive enough to report small subpopulation of vaible bacteria in the cultures that typically survive antibiotic exposure. Therefore, plating out the dilutions of biofilms will be critical to evaluate the extent of sterilization by vitamin C. The enthusiasm of this reviewer is further dampened by the fact that vitamin C has not been tested in the in vivo model of Mtb growth, and therefore questions remain if vitamin C will at all be useful in therapeutics.

Validity of the findings

As I have noted earlier, additional assays are necessary to determine the extent of mycobacterial killing in biofilms by Vitamin C. Luciferase assay is not sensitive enough to determine the extent of sterilization of biofilms by Vitamin C. Moreover, including anti-TB drugs in parallel experiments could be critical in making a comparisons, that authors intend to make in the paper.

Additional comments

Overall, I think this paper has some value as it is extending the previous work of Jacobs and colleagues in evaluating the anti-TB activity of vitamin C against Mtb biofilms. However, the experimental design is too preliminary.

·

Basic reporting

Reference number 3 (line 163) appears to be part of reference number 2. Therefore all text references beginning with #3 (line 43) actually correspond to one number higher in the reference section.

Experimental design

This study would benefit enormously by including other treatments besides ascorbic acid., notably existing and experimental anti-TB drugs (rifampin, bedaquiline, nitroimidazoles, fluoroquinolones, aminoglycosides, INH, ethambutol, etc.) and possibly compounds more related to ascorbic acid. This would place the ascorbic acid results in better context.

Is there a reasonable possibility that the use of a glycerol- based medium (7H9 + ADC + glycerol) to assess cidality might miss a phenotype which will only grow out on a fatty acid-based medium (e.g. 7H9 + OADC, or 7H12 medium)? For that matter, considering that glycerol is considered to be an irrelevant carbon source in vivo, is there a reason that glycerol-containing media were used to produce the biofilm and to assess the activity of ascorbic acid, especially considering the experience of NITD in pursuing (what turned out to be) a glycerol-dependent hit.

Validity of the findings

No comments

Additional comments

The last part of the abstract (lines 28-30) refers to the potential value of novel antibiotics acting by a similar mechanism of action of ascorbic acid but there is no expansion at all of this concept in the discussion. This would make the paper more interesting.

The first question that comes to mind (at least for me) is the relevance of the effective concentrations. There is an excellent review (M. Levine et al., Adv. Nutr. 2:78-88, 2011) indicating that such concentrations are achievable in vivo (only) when administering IV. I would suggest including this reference.

Reviewer 3 ·

Basic reporting

This is an interesting, brief report. The question it addresses – whether the sensitivity of M.tb to ascorbic acid is similar in biofilms as it is to cultures grown in planktonic phase – is an important one. I found that the way this question was framed was misleading. The authors state that “many… phenotypes and niches [found during natural infection] are replicated within a biofilm”. There is no citation to support this statement. I am not aware of any evidence that this is the case for M.tb and, indeed, this would be extremely difficult to demonstrate. They later assert that M.tb’s growth in pellicles “correlates better to the disease environment”. What is the basis for making such a statement? The authors are on much firmer ground in asserting that it is more difficult to sterilize biofilms than bacteria in planktonic growth, and that biofilms contain a mixture of “replicating and non-replicating cells” although a citation is needed here as well.

Some of the citations are incorrect. For example, the authors cite a study of A. baumannii in an insect model to support the statement that mycobacterial species form complex, secondary structures in vitro (citation 14). This may be a numbering issue.

Experimental design

The experimental design is sound and clearly described. One question I have – how did the authors determine the amount of media remaining in the wells at the end of the 8 week period?

Validity of the findings

The difference between cultures that received >= 32 mM ascorbic acid, and those that did not, seem clear. Cultures exposed to the high concentration had light levels lower than observed at day 0 (dotted lines on Fig 1), whereas those with a low concentration had similar/higher levels. However, light levels on day 0 were presumably measured for a number of different wells/cultures. How variable were they? What value was chosen to show on Fig 1 and why?

In the assessment of viability after ascorbic acid treatment (Fig 2), there are some details missing. What proportion of the untreated/low treatment cultures demonstrated growth during this stage? It would also be helpful to include low ascorbic acid data in Fig 2 (there are no points between 0 and 32mM).

The authors cite their own unpublished data regarding efficacy of “other drugs” against M.tb pellicles. More detail is needed here. Did they compare efficacy of drugs against M.tb grown on surfaces versus cultures grown as pellicles? Or did they compare planktonic with pellicle growth? What drugs were tested?

---

## Round 0.2 · accepted · Accept

In the absence of reaction from the original reviewers, I took it on myself to examine your rebuttal. Although not every point was accepted, it seems that most of the critical corrections have been made.